# RELEVANCE ATTACK ON DETECTORS

## ABSTRACT

This paper focuses on high-transferable adversarial attacks on detectors, which are hard to attack in a black-box manner, because of their multiple-output characteristics and the diversity across architectures. To pursue a high attack transferability, one plausible way is to find a common property across detectors, which facilitates the discovery of common weaknesses. We are the first to suggest that the relevance map for detectors is such a property. Based on it, we design a Relevance Attack on Detectors (RAD), which achieves a state-of-the-art transferability, exceeding existing results by above 20%. On MS COCO, the detection mAPs for all 8 black-box architectures are more than halved and the segmentation mAPs are also significantly influenced. Given the great transferability of RAD, we generate the first adversarial dataset for object detection, i.e., Adversarial Objects in COntext (AOCO), which helps to quickly evaluate and improve the robustness of detectors.

## 1 INTRODUCTION

Adversarial attacks (Szegedy et al. (2014); Goodfellow et al. (2015); Carlini & Wagner (2017); Mądry et al. (2017); Baluja & Fischer (2017); Su et al. (2019)) have revealed the fragility of Deep Neural Networks (DNNs) by fooling them with elaborately-crafted imperceptible perturbations. Among them, the black-box attack, i.e., attacking without knowledge of their inner structure and weights, is much harder, more aggressive and closer to real-world scenarios. For classifiers, there exist some promising black-box attacks (Papernot et al. (2016); Brendel et al. (2018); Dong et al. (2018); Xie et al. (2019); Lin et al. (2020); Chen et al. (2020)). It is also severe to attack object detection (Zhang & Wang (2019)) in a black-box manner, e.g., hiding certain objects from unknown detectors (Thys et al. (2019)). By that, life-concerning systems based on detection such as autonomous driving and security surveillance would be greatly influence.

To the best of our knowledge, no existing attack is specifically designed for black-box transferability in detectors, because they have multiple-outputs and a high diversity across architectures. In such situations, adversarial samples do not transfer well (Su et al. (2018)), and most attacks only decrease mAP of black-box detectors by 5 to 10% (Xie et al. (2017); Li et al. (2018c;b)). To overcome this, we propose one plausible way to find common properties across detectors, which facilitates the discovery of common weaknesses. Based on them, the designed attack can threaten variable victims.

In this paper, we adopt the relevance map as a common property, on which different detectors have similar interpretable results, as shown in Fig. 1. Based on relevance maps, we design a Relevance Attack on Detectors (RAD). RAD focuses on suppressing the relevance map rather than directly attacking the prediction as in existing works (Xie et al. (2017); Li et al. (2018c;a)). Because the relevance maps are quite similar across models, those of black-box models are influenced and misled as well in attack, leading to the great transferability. Although some works have adopted the relevance map as an indicator or reference of success attacks (Dong et al. (2019); Zhang & Zhu (2019); Chen et al. (2020); Wu et al. (2020a)), there is no work to directly attack the relevance maps of detectors to the best of our knowledge.

In our comprehensive evaluation, RAD achieves the state-of-the-art transferability on 8 black-box models for COCO) dataset (Lin et al. (2014)), nearly halving the detection mAP. Interestingly, the adversarial samples of RAD also greatly influence the performance of instance segmentation, even only detectors are attacked. Given the high transferability of RAD, we create Adversarial Objects in COntext (AOCO), the first adversarial dataset for object detection. AOCO contains 10K samples that significantly decrease the performance of black-box models for detection and segmentation. AOCO may serve as a benchmark to test the robustness of a DNN or improve it by adversarial training.

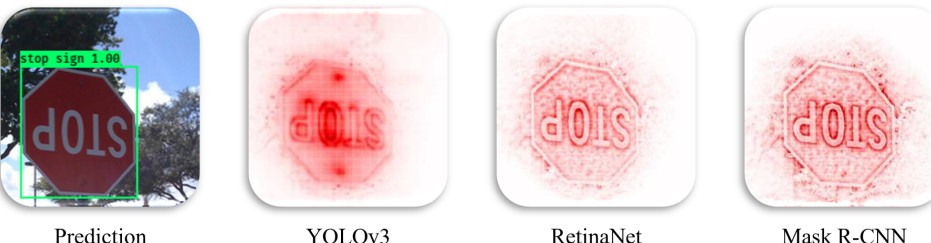

| Prediction | YOLOv3 | RetinaNet | Mask R-CNN |

Figure 1: Relevance maps for models with different architectures. Three models not only predict the "stop sign" right, but also share similar relevance maps.

CONTRIBUTIONS

- We propose a novel attack framework on relevance maps for detectors. We extend network visualization methods to detectors, find out the most suitable nodes to attack by relevance maps, and explore on the best update techniques to increase the transferability.

- We evaluate RAD comprehensively and find its state-of-the-art transferability, which exceeds existing results by above 20%. Detection mAPs are more than halved, invalidating the state-of-the-art detectors to a large extent.

- By RAD, we create the first adversarial dataset for object detection, i.e., AOCO. As a potential benchmark, AOCO is generated from COCO and contains 10K high-transferable samples. AOCO helps to quickly evaluate and improve the robustness of detectors.

## 2 RELATED WORK

Since (Szegedy et al. (2014)), there have been lots of promising adversarial attacks (Goodfellow et al. (2015); Carlini & Wagner (2017); Mądry et al. (2017)). Generally, they fix the network weights and change the input slightly to optimize the attack loss. The network then predicts incorrectly on adversarial samples with a high confidence. (Papernot et al. (2016; 2017)) find that adversarial samples crafted by attacking a white-box surrogate model may transfer to other black-box models as well. Input modification (Xie et al. (2019); Dong et al. (2019); Lin et al. (2020)) or other optimization ways (Dong et al. (2018); Lin et al. (2020)) are validated to be effective in enhancing the transferability.

(Xie et al. (2017)) extends adversarial attacks to detectors. It proposes to attack on densely generated bounding boxes. After that, losses about localization and classification are designed (Li et al. (2018c)) for attacking detectors. (Lu et al. (2017)) and (Li et al. (2018a)) propose to attack detectors in a restricted area. Existing works achieve good results in white-box scenarios, but are not specifically designed for transferability. The adversarial impact on black-box models is quite limited, i.e., a 5 to 10% decrease from the original mAP, even when two models only differ in backbone (Xie et al. (2017); Li et al. (2018c;b)). (Wang et al. (2020)) discusses black-box attacks towards detectors based on queries rather than the transferability as we do. The performance is satisfactory, but it requires over 30K queries, which is easy to be discovered by the model owner. Besides, physical attacks on white-box detectors are also feasible (Huang et al. (2020); Wu et al. (2020b); Xu et al. (2020)).

For the great transferability, we propose to attack on relevance maps, which are calculated by network visualization methods (Zeiler & Fergus (2014); Selvaraju et al. (2017); Shrikumar et al. (2017)). They are originally developed to interpret how DNNs predict and help users gain trust on them. Specifically, they display how the input contributes to a certain node output in a pixel-wise manner. Typical works include Layer-wise Relevance Propagation (LRP) (Bach et al. (2015)), Contrastive LRP (Gu et al. (2018)) and Softmax Gradient LRP (SGLRP) (Iwana et al. (2019)). These methods encourage the reference of relevance maps in attack (Dong et al. (2019); Zhang & Zhu (2019); Chen et al. (2020); Wu et al. (2020a)), and also inspire us. However, none of them attack on relevance maps for detectors.

RAD differs from (Ghorbani et al. (2019); Zhang et al. (2020)) in the goal. RAD misleads detectors by suppressing relevance maps. In contrast, (Ghorbani et al. (2019)) misleads the relevance maps while keeping the prediction unchanged. (Zhang et al. (2020)) also misleads DNNs, but it keeps the relevance maps unchanged.

## 3 Relevance Attack on Detectors

We propose an attack specifically designed for black-box transferability, named Relevance Attack on Detectors (RAD). RAD suppresses multi-node relevance maps for several bounding boxes. Since the relevance map is commonly shared by different detectors as shown in Fig. 1, attacking on it in the white-box surrogate model achieves a high transferability towards black-box models. In this section, we first provide a high-level overview of RAD, and analyze the potential reasons of its transferability. Then we thoroughly discuss three crucial concrete issues in RAD.

- In Section 3.3, we specify the calculation of relevance maps for detectors, where current visualization methods are not applicable.
- In Section 3.4, we introduce the proper nodes to attack by RAD.
- In Section 3.5, we explore on the suitable techniques to update samples in RAD.

### 3.1 What is RAD?

We present the framework of RAD in Fig. 2. Initialized by the original sample $x_0$, the adversarial sample $x_k$ in the $k^{\text{th}}$ iteration is forward propagated in the surrogate model, getting the prediction $f(x_k)$. Current attacks generally suppress the prediction values of all attacked output nodes in $T$. In contrast, RAD suppresses the corresponding relevance map $h(x_k, T)$. To restrain that, gradients of $h(x_k, T)$ back propagate to $x_k$, which is then modified to $x_{k+1}$.

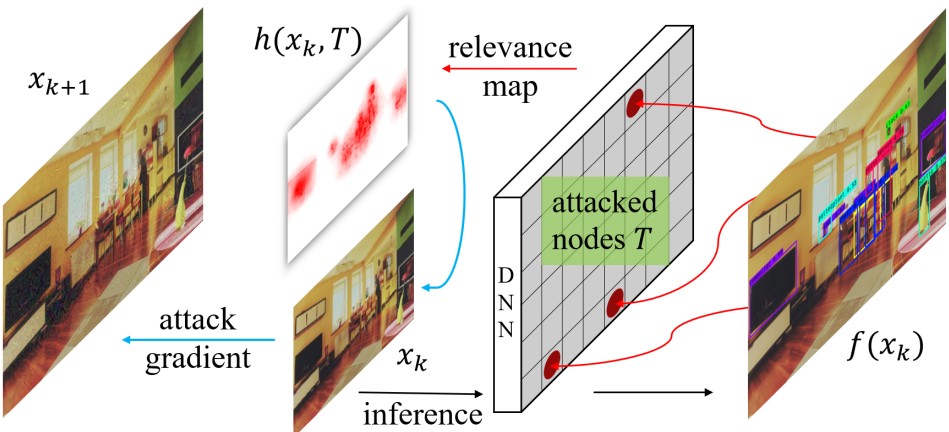

Figure 2: Framework of RAD. $x_k$ is the sample in iteration $k$ and $f(x_k)$ is the network prediction for it. $h(x_k, T)$ stands for the relevance map for all attacked nodes in $T$. RAD works by repeating processes denoted by "black", "red" and "blue" arrows in turn.

It is notable that RAD is a complete framework to attack detectors, and its each component requires a special design. Besides the calculation of relevance maps of detectors, other components in RAD, e.g., the attacked nodes or the update techniques, also need a customized analysis. The reason is that no existing work directly attacks the relevance of detectors, and the experience in attacking predictions is not totally applicable here. For example, (Zhang & Wang (2019)) emphasizes classification loss and localization loss equally, but the former is validated to be significantly better in attacking the relevance in Section 3.4.

### 3.2 Why RAD Transfers?

RAD's transferability comes from the attack goal: changing the common properties, i.e., the relevance maps. As shown in Fig. 3, the relevance maps are clear and structured for the original sample in both detectors. After RAD, the relevance maps are induced to be meaningless without a correct focus, leading to wrong predictions, i.e., no or false detection. Because relevance maps transfer well across models, those for black-box detectors are also significantly influenced, causing a great performance drop, which is illustrated visually in Section 4.2.

RetinaNet              Mask R-CNN

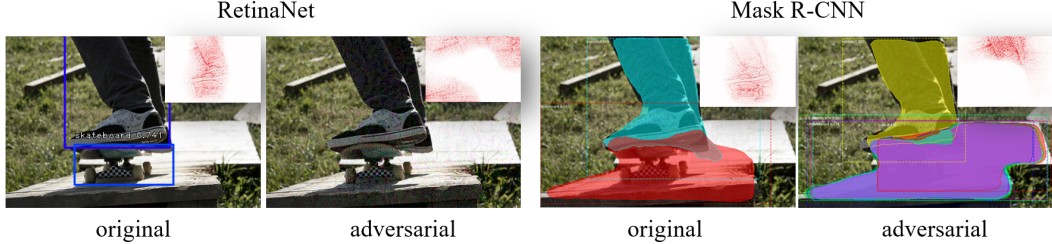

original     adversarial     original     adversarial

Figure 3: RAD's transferability origins from the change of relevance maps. The image contains a person and a skateboard. By attacking on relevance maps, both surrogate models make extremely confusing predictions.

RAD also attacks quite "precisely", i.e., the perturbation pattern is significantly focused on distinct areas and has a clear structure as shown in Fig. 4. That is to say, RAD accurately locates the most discriminating parts of a sample and concentrates the perturbation on them, leading to a great transferability when the perturbations is equally bounded.

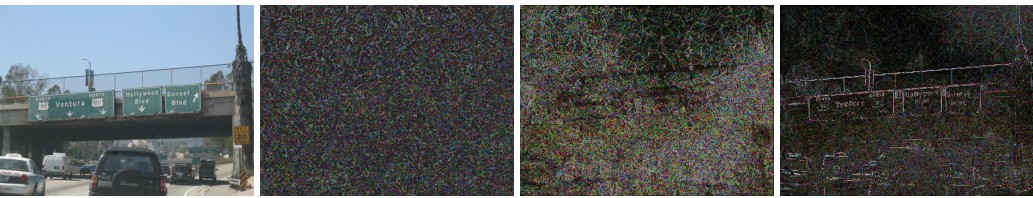

Figure 4: The original image and the adversarial perturbations ($\times 5$ in magnitude for demonstration) generated by Dfool (Lu et al. (2017)), DAG (Xie et al. (2017)), and RAD (from left to right)

### 3.3 What is the Relevance Maps for Detectors?

We analyze the potential of RAD above, below we make it feasible by addressing three crucial issues. To conduct the relevance attack, we first need to know the relevance maps for detectors.

Currently, there have been lots of methods to calculate the relevance maps for classifiers as described in Section 2, but none of them are suitable for detectors. We take SGLRP (Iwana et al. (2019)) as an example to explain this and then to modify, because it excels in discriminating ability against irrelevant regions of a certain target node.

SGLRP visualizes how the input contributes to one output node in a pixel-wise way by back-propagating the relevance from the output to the input based on Deep Taylor Decomposition (DTD) as illustrated in Appendix A. $R^{(L)}$ is the initial relevance in the output layer $L$ and its $n^{\text{th}}$ component is calculated as

$$R_n^{(L)} = \begin{cases} y_n\left(1 - y_n\right) & n = t, \\ -y_n y_t & n \neq t, \end{cases} \tag{1}$$

where $y_n$ is the predicted probability of class $n$, and $y_t$ is that for the single-node target $t$. The pixel-wise relevance map $h(x, t)$ for the single-node target $t$ is calculated by back propagating the relevance $R$ from the final layer to the input following rules specified in (Iwana et al. (2019)).

In detectors, we need the pixel-wise contributions from the input to $m$ bounding boxes. This multi-node relevance map could not be directly calculated by (1), so we naturally modify SGLRP as

$$R_n^{(L)} = \begin{cases} y_n\left(1 - y_n\right) & n \in T, \\ -\frac{1}{m} y_n \sum_{i=1}^m y_{t_i} & n \notin T, \end{cases} \tag{2}$$

where $y_{t_i}$ is the predicted probabilities for one target node $t_i$. $T$ is the set containing all target nodes $\{t_1, t_2, ..., t_m\}$. With iNNvestigate Library (Alber et al. (2019)) to implement Multi-Node SGLRP and Deep Learning platforms supporting auto-gradient, the gradients from RAD loss $L_{\text{RAD}}(x) = h(x, T)$ to sample $x$ could be obtained according to the calculation rules of relevance maps in Appendix A.

We illustrate the difference between SGLRP and our Multi-Node SGLRP in Fig. 5. SGLRP only displays the relevance map for one bounding box, e.g., "TV", "chair" and "bottle". Multi- Node SGLRP, in contrast, visualizes the overall relevance.

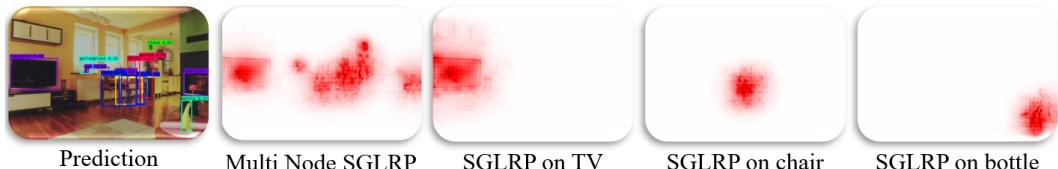

| Prediction | Multi Node SGLRP | SGLRP on TV | SGLRP on chair | SGLRP on bottle |

Figure 5: Difference between relevance maps from SGLRP and Multi-Node SGLRP. The relevance maps are for YOLOv3 (Redmon & Farhadi (2018)).

### 3.4 WHERE TO ATTACK?

Besides the calculation of relevance maps, it is also important to choose a proper node set $T$ to attack. Specifically, we need to select certain bounding boxes and the corresponding output nodes for RAD.

Heuristically, the most "obvious" bounding boxes are desired to be eliminated, so we select the bounding boxes with the highest confidence, following (Xie et al. (2017)). Concretely, it is feasible to *statically* choose $m$ bounding boxes to attack in each iteration, or *dynamically* attack all bounding boxes whose confidence exceeds a threshold. In our evaluation, the two strategies differ a little in performance and are not sensitive to hyper-parameter as demonstrated in Appendix C. This shows that RAD does not require a sophisticated tuning of parameters, which is user-friendly. In our following experiments, we statically attack $m = 20$ nodes.

After selecting bounding boxes, we could attack their size, leading them to shrink; or their localization, leading them to shift; or their confidence, leading them to be misclassified. To adopt the best strategy, we conduct a toy experiment by attacking YOLOv3 (Redmon & Farhadi (2018)), denoted as M2 and other models are specified in Appendix B. Given the results in Table 1, the classification loss induces a better black-box transferability. This may because detectors generally include a pre-trained classification as the feature extractor, and relevance maps are believed to be an indicator of success attacks (Dong et al. (2019); Zhang & Zhu (2019)).

Table 1: Detection mAP in RAD with different attacked nodes

| Strategy | M1 | M2 | M3 | M4 | M5 | M6 | M7 | M8 | M9 |
|---|---|---|---|---|---|---|---|---|---|
| No Attack | 29.3 | 33.4 | 38.1 | 40.7 | 42.1 | 42.5 | 45.7 | 46.9 | 53.9 |
| Size | 26.0 | 14.7 | 31.9 | 32.5 | 35.6 | 35.4 | 38.6 | 40.0 | 47.8 |
| Localization | 22.8 | 6.4 | 27.4 | 28.1 | 31.7 | 30.8 | 34.4 | 35.9 | 45.1 |
| Classification | **18.1** | **1.2** | **19.9** | **20.5** | **24.3** | **22.6** | **26.4** | **28.2** | **39.9** |

### 3.5 HOW TO UPDATE?

By the relevance map $h(x, T)$ for certain attacked nodes $T$, we are able to attack, i.e., update the original sample to become adversarial with the guidance of the attack gradients $g(x)$ as

$$g(x) = \frac{\partial L_{\text{RAD}}(x)}{\partial x} = \frac{\partial h(x, T)}{\partial x}. \tag{3}$$

Some update techniques are validated to be effective for enhancing the transferability in classification. For example, Scale-Invariant (SI) (Lin et al. (2020)) proposes to average the attack gradients by scale copies of the samples as

$$g_{\text{si}}(x) = \frac{1}{k} \sum_{i=0}^{k} g(x/2^i). \tag{4}$$

Besides SI, Diverse Input (DI) (Xie et al. (2019)), Translation-Invariant (TI) (Dong et al. (2019)) are also promising in classification. We are curious about whether they also work well in object detection. To explore on this, we adopt these techniques in RAD as the setting suggested by their designers (see Appendix E). From the results in Table 2, we discover that SI is quite effective, further decreasing the mAP from the baseline to a large extent. Accordingly, we adopt (4) to update the sample in RAD.

Table 2: Detection mAP in RAD with different update techniques

| Technique | M1 | M2 | M3 | M4 | M5 | M6 | M7 | M8 | M9 |
|---|---|---|---|---|---|---|---|---|---|
| None | 18.1 | 1.2 | 19.9 | 20.5 | 24.3 | 22.6 | 26.4 | 28.2 | 39.9 |
| DI | 18.1 | 1.0 | 19.9 | 20.5 | 23.9 | 22.4 | 26.3 | 27.9 | 39.6 |
| TI | 17.0 | 2.4 | 20.8 | 20.8 | 25.2 | 23.0 | 27.9 | 29.7 | 41.5 |
| SI | **14.6** | **0.7** | **16.3** | **17.0** | **20.4** | **19.1** | **22.3** | **23.8** | **35.0** |

With the calculated gradient, we update the sample as

$$x_{k+1} = \text{clip}_\varepsilon \left( x_k - \alpha \frac{g_{\text{si}}(x_k)}{||g_{\text{si}}(x_k)||_1/N} \right), \tag{5}$$

where $\alpha$ stands for the step length. $x$ is $\ell_\infty$-norm bounded by $\varepsilon$ from the original sample in each iteration as in (Xie et al. (2019); Dong et al. (2019); Lin et al. (2020)). Gradient $g(x)$ is normalized by its average $\ell_1$-norm,i.e., $||g(x)||_1/N$ to prevent numerical errors and control the degree of perturbations. $N$ is the dimension of the image, i.e., $N = height \times width \times channel$. Division by $N$ is necessary because $\ell_1$-norm sums all components of the tensor $x$, which is too large as a normalization factor. We do not adopt the mainstream *sign* method because it is not suitable to generate small perturbations as shown in other attacks in detectors (Xie et al. (2017)).

## 4    EXPERIMENTS

In this section, we evaluate the performance of RAD, especially its transferability. The results are presented numerically and visually. In comprehensive evaluation, RAD achieves a great transferability in across models and even across tasks.

### 4.1    SETUP

Our experiments are based on Keras (Chollet et al. (2015)), Tensorflow (Abadi et al. (2015)) and PyTorch (Paszke et al. (2019)) in 4 NVIDIA GeForce RTX 2080Ti GPUs. Library iNNvestigate (Alber et al. (2019)) is used to implement Multi-Node SGLRP.

We conduct experiments on MS COCO 2017 dataset (Lin et al. (2014)), which is a large-scale benchmark for object detection, instance segmentation and image captioning. For a fair evaluation, we generate adversarial samples from all 5K samples in its validation set and test several black-box models on their mAP, a standard criteria in many works (He et al. (2017); Chen et al. (2019a)).

All attacks are conducted with the step length $\alpha = 2$ for 10 iterations and the perturbation is $\ell_\infty$-bounded in $\varepsilon = 16$ to guarantee the imperceptibility as in (Dong et al. (2019)). To validate that the mAP drop comes from the attack instead of resizing or perturbation, we add large Gaussian noises ($\sigma = 9$) to the resized images, and report it as "Ablation".

We choose 8 typical detectors ranging from the first end-to-end detector to recent ones for attack and test. The variety of model guarantees the validity of results. We specify their information in Appendix B and the corresponding pre-processing in Appendix E.

### 4.2    VISUAL RESULTS OF RAD

We visualize several predictions on the same adversarial sample by black-box models in Fig. 6 to intuitively illustrate the transferability of RAD. The objects in the image, e.g., the laptop and keyboard, are quite large and obvious to detect. However, with a small perturbation from RAD, 5 black-box models all fail to detect the laptop, keyboard and mouse. Surprisingly, 4 of them even detect a non-existent "bed", which is neither relevant nor similar in the image. The attack process of RAD is analyzed in Appendix F.

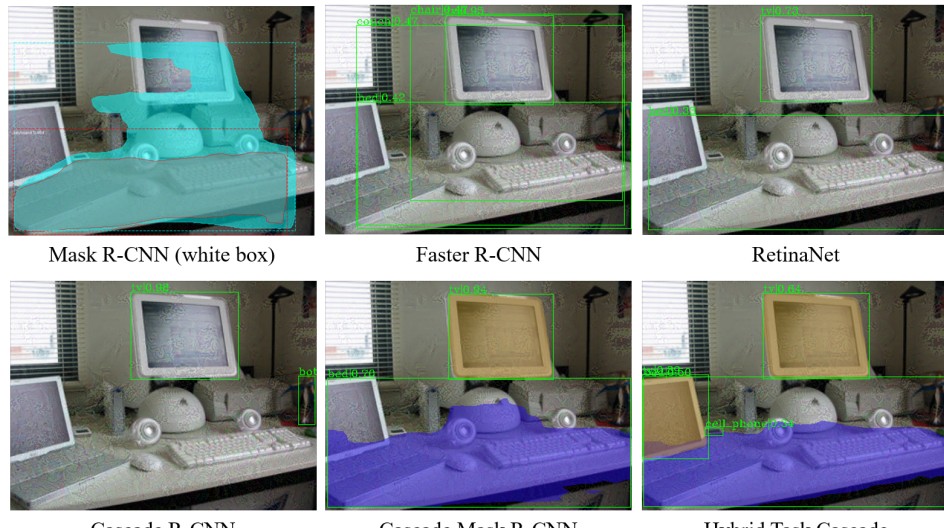

Mask R-CNN (white box)          Faster R-CNN          RetinaNet

Cascade R-CNN          Cascade Mask R-CNN          Hybrid Task Cascade

Figure 6: RAD has a great transferability. The same adversarial sample generated by attacking Mask R-CNN fools all 5 black-box detectors.

### 4.3 RAD's Transferability in Object Detection

To evaluate the in-domain transferability of detection attacks and cross-domain transferability of classification attacks, we test the detection mAP of 8 models in COCO adversarial samples generated in the same setting.

For detection attacks, adversarial samples are crafted by attacking surrogate model M2 (YOLOv3 Redmon & Farhadi (2018)). Results of attacking other surrogates are reported in Appendix D. For classification attacks, we use the model output on the clean sample as the label. By several state-of-the-art attacks on surrogate classifiers (InceptionV3 Szegedy et al. (2016) here as in Xie et al. (2019); Dong et al. (2019)), the adversarial samples are generated and tested the mAP as the transferability towards detectors. Details of implementation are described in Appendix E.

We present the results in Table 3. Among the classification attacks and detection ones, cross-domain attack (Naseer et al. (2019)) is effective, but RAD is more aggressive. RAD enjoys a state-of-the-art transferability towards most black-box models, outperforming other methods for above 20%. The detection mAPs are more than halved, making state-of-the-art detectors worse than the early single-shot detector (SSD Liu et al. (2016), M1).

Table 3: Detection mAP in different attacks

|  | Method | M1 | M2 | M3 | M4 | M5 | M6 | M7 | M8 | M9 |
|---|---|---|---|---|---|---|---|---|---|---|
| Reference | No Attack | 29.3 | 33.4 | 38.1 | 40.7 | 42.1 | 42.5 | 45.7 | 46.9 | 53.9 |
|  | Ablation | 24.9 | 31.4 | 31.2 | 31.6 | 35.0 | 34.3 | 37.5 | 38.8 | 48.6 |
| Classification Attack | PGD | 26.4 | 30.4 | 34.4 | 35.4 | 38.4 | 38.3 | 41.7 | 43.1 | 51.1 |
|  | SI-PGD | 27.5 | 31.6 | 36.1 | 37.1 | 40.0 | 40.1 | 43.5 | 44.8 | 52.4 |
|  | MI-DI-PGD | 22.9 | 26.2 | 29.3 | 30.0 | 33.2 | 32.1 | 36.0 | 37.5 | 48.0 |
|  | MI-TI-PGD | 20.1 | 23.7 | 24.9 | 25.4 | 30.1 | 27.4 | 32.8 | 34.5 | 47.1 |
|  | CD-painting | 16.4 | 20.8 | 21.3 | 22.8 | 26.6 | 24.5 | 28.9 | 29.5 | 42.3 |
|  | CD-comics | 16.6 | 21.6 | 21.7 | 22.7 | 26.8 | 24.3 | 29.1 | 42.3 | 43.7 |
| Detection Attack | Dfool | 23.3 | 2.5 | 29.2 | 29.8 | 33.3 | 32.9 | 36.5 | 38.0 | 47.5 |
|  | Loc | 21.9 | **0.2** | 25.8 | 26.6 | 29.8 | 29.4 | 33.2 | 33.2 | 45.2 |
|  | DAG | 20.8 | 0.6 | 22.8 | 23.4 | 26.8 | 25.6 | 28.9 | 31.0 | 40.6 |
| Ours | RAD | **14.6** | 0.7 | **16.3** | **17.0** | **20.4** | **19.1** | **22.3** | **23.8** | **35.0** |

The influence of $\varepsilon$ on detection mAP in RAD are displayed in Fig. 7. With the $\ell_\infty$ bound increases, the resulting mAP greatly decreases for all black-box models especially for $\varepsilon$ from 8 to 12.

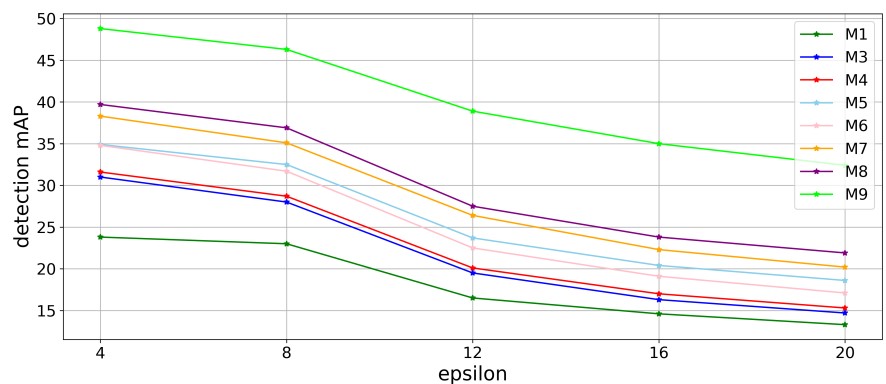

Figure 7: The influence of $\varepsilon$ on detection mAP in RAD

## 4.4 RAD'S TRANSFERABILITY TO INSTANCE SEGMENTATION

Detection and segmentation are similar in some aspects, so they could be implemented in one network (He et al. (2017); Cai & Vasconcelos (2018); Chen et al. (2019a)). Also, adversarial samples for object detection tend to transfer to instance segmentation (Xie et al. (2017)). Accordingly, we evaluate this cross-task transferability by RAD on surrogate detectors YOLOv3 (Redmon & Farhadi (2018), M2), RetinaNet (Lin et al. (2017), M3) and Mask R-CNN (He et al. (2017), M5). From the results in Table 4, we find that RAD also greatly hurts the performance of instance segmentation, leading to a drop on mAP of over 70%. This inspire the segmentation attackers to indirectly attack detectors.

Table 4: Segmentation mAP of RAD

| Surrogate | mAP | | | mAP50 | | | mAP75 | | |
|---|---|---|---|---|---|---|---|---|---|
| | M5 | M7 | M8 | M5 | M7 | M8 | M5 | M7 | M8 |
| None | 38.0 | 39.4 | 40.8 | 60.6 | 61.3 | 63.3 | 40.9 | 42.9 | 44.1 |
| Ablation | 31.0 | 31.9 | 33.5 | 51.2 | 51.0 | 53.7 | 32.4 | 34.3 | 35.4 |
| M2 | 17.9 | 18.6 | 20.3 | 31.6 | 31.7 | 34.5 | 18.0 | 18.9 | 20.7 |
| M3 | 11.6 | 11.9 | 12.9 | 19.2 | 19.1 | 20.7 | 12.1 | 12.6 | 13.7 |
| M5 | 1.2 | 11.1 | 11.8 | 2.4 | 17.9 | 18.9 | 1.0 | 11.9 | 12.6 |

## 5 ADVERSARIAL OBJECTS IN CONTEXT

Given the great transferability of RAD, we create Adversarial Objects in COntext (AOCO), the first adversarial dataset for object detection. AOCO dataset serves as a potential benchmark to evaluate the robustness of detectors, which will be beneficial to network designers. It will also be useful for adversarial training, as the most effective practice to improve the robustness of DNNs Zhang et al. (2019); Tramèr et al. (2018). Notice that there is no other adversarial dataset for detection at all. This is not because the dataset is useless, but due to the low transferability of attack methods such that the examples are detector-dependent. Now we have achieved high transferability and can then make such an adversarial dataset publicly available.

AOCO is generated from the full COCO 2017 validation set (Lin et al. (2014)) with 5k samples. It contains 5K adversarial samples for evaluating object detection (AOCO detection) and 5K for instance segmentation (AOCO segmentation). All 10K samples in AOCO are crafted by RAD. The surrogate

model we attack is YOLOv3 for AOCO detection and Mask R-CNN for AOCO segmentation given the results in Table 3 and Table 4.

We measure the perturbation $\Delta x$ in AOCO by Root Mean Squared Error (RMSE) as in (Xie et al. (2017); Liu et al. (2017)). It is calculated as $\sqrt{\sum_i (\Delta x_i)^2 / N}$ in a pixel-wise way, and $N$ is the size of the image. Performance of AOCO is reported in Table 5. The RMSE in AOCO is lower than that in (Wu et al. (2019)), and the perturbation is quite imperceptible. Details and samples of AOCO are presented in Appendix G.

Table 5: Detection mAP and segmentation mAP on COCO and AOCO

|  | RMSE | M1 | M2 | M3 | M4 | M5 | M6 | M7 | M8 | M9 |
|---|---|---|---|---|---|---|---|---|---|---|
| COCO detection | 0.000 | 29.3 | 33.4 | 38.1 | 40.7 | 42.1 | 42.5 | 45.7 | 46.9 | 53.9 |
| AOCO detection | 6.469 | 14.6 | 0.7 | 16.3 | 17 | 20.4 | 19.1 | 22.3 | 23.8 | 35.0 |
| COCO segmentation | 0.000 | \ | \ | \ | \ | 38.0 | \ | 39.4 | 40.8 | \ |
| AOCO segmentation | 6.606 | \ | \ | \ | \ | 1.2 | \ | 11.1 | 11.8 | \ |

## 6 CONCLUSION

To pursue a high transferability, this paper proposes Relevance Attack on Detectors (RAD), which works by suppressing the multi-node relevance, a common property across detectors calculated by our Multi-Node SGLRP. We also thoroughly discuss where to attack and the how to update in attacking relevance maps. RAD achieves a state-of-the-art transferability towards 8 diverse black-box models, exceeding existing results by above 20%, and also significantly hurts the instance segmentation. Given the great transferability of RAD, we generate the first adversarial dataset for object detection, i.e., Adversarial Objects in COntext (AOCO), which helps to quickly evaluate and improve the robustness of detectors. Also, attacking other common properties is promising for a good transferability.

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

## A    RELEVANCE BACK-PROPAGATION RULES (DTD)

DTD-based network visualization methods, such as LRP, CLRP and SGLRP, back-propagate the relevance from the output layer to the input layer according to the rules specified in this section. Their only difference is the relevance in the initial output layer $R_n^{(L)}$.

For each layer $l$ in a DNN with $L$ layers in total, suppose layer $l$ has $N$ nodes and layer $l + 1$ has $M$ nodes, the relevance $R_n^{(L)}$ at node $n$ in layer $l$ is defined recursively by

$$R_n^{(l)} = \sum_m \frac{a_n^{(l)} w_{n,m}^{+(l)}}{\sum_{n'} a_{n'}^{(l)} w_{n',m}^{+(l)}} R_m^{(l+1)},$$

for nodes with definite positive values (such as after ReLU), and

$$R_n^{(l)} = \sum_m \frac{z_n^{(l)} w_{n,m}^{(l)} - b_n^{(l)} w_{n,m}^{+(l)} - h_n^{(l)} w_{n,m}^{-(l)}}{\sum_{n'} z_{n'}^{(l)} w_{n',m}^{(l)} - b_{n'}^{(l)} w_{n',m}^{+(l)} - h_{n'}^{(l)} w_{n',m}^{-(l)}} R_m^{(l+1)},$$

for nodes that may have negative values. In the formulas above, $a_n^{(l)}$ is the post-activation output of node $n$ in layer $l$ and $z_n^{(l)}$ is the pre-activation one. The range $[b_n^{(l)}, h_n^{(l)}]$ stands for the minimum and maximum of $z_n^{(l)}$. Finally, $w_{n,m}^{+(l)} = \max\left(w_{n,m}^{(l)}, 0\right)$ and $w_{n,m}^{-(l)} = \min\left(w_{n,m}^{(l)}, 0\right)$.

According to the propagation rules above as mentioned in (Iwana et al. (2019)), we could naturally obtain the attack gradients as

$$\frac{\partial R_m^{(l+1)}}{\partial R_n^{(l)}} = \begin{cases} \left(\sum_m \frac{a_n^{(l)} w_{n,m}^{+(l)}}{\sum_{n'} a_{n'}^{(l)} w_{n',m}^{+(l)}}\right)^{-1}, \text{for nodes with definite positive values} \\ \left(\sum_m \frac{z_n^{(l)} w_{n,m}^{(l)} - b_n^{(l)} w_{n,m}^{+(l)} - h_n^{(l)} w_{n,m}^{-(l)}}{\sum_{n'} z_{n'}^{(l)} w_{n',m}^{(l)} - b_{n'}^{(l)} w_{n',m}^{+(l)} - h_{n'}^{(l)} w_{n',m}^{-(l)}}\right)^{-1}, \text{otherwise} \end{cases}.$$

## B    MODEL INFORMATION

Table 6 presents the models' information from our evaluation and MMdetection (Chen et al. (2019b)).

Table 6: Model backbone and mAPs

| ID | Model | Backbone | mAP |
|---|---|---|---|
| M1 | SSD512 (Liu et al. (2016)) | VGG16 | 29.3 |
| M2 | YOLOv3 (Redmon & Farhadi (2018)) | Darknet | 33.4 |
| M3 | RetinaNet (Lin et al. (2017)) | ResNet-101 | 38.1 |
| M4 | Faster R-CNN (Ren et al. (2015)) | ResNeXt-101-64*4d | 40.7 |
| M5 | Mask R-CNN (He et al. (2017)) | ResNeXt-101-64*4d | 42.1 |
| M6 | Cascade RCNN (Cai & Vasconcelos (2018)) | ResNet-101 | 42.5 |
| M7 | Cascade Mask R-CNN (Cai & Vasconcelos (2018)) | ResNeXt-101-64*4d | 45.7 |
| M8 | Hyrbrid Task Cascade (Chen et al. (2019a)) | ResNeXt-101-64*4d | 46.9 |
| M9 | EfficientDet (Tan et al. (2020)) | EfficientNet + BiFPN | 53.9 |

## C    INFLUENCE OF HYPER-PARAMETERS IN NODE SELECTION

Performance of RAD is not sensitive to hyper-parameter, no matter the strategy to select bounding boxes is dynamic or static as Table 7. Attackers are not bothered to tune them carefully. The parameter for dynamic strategy refers to the pre-softmax confidence threshold to select a bounding box. The parameter for static strategy refers to the fixed number of selected bounding boxes in each iteration.

## D    RAD ON MORE SURROGATES

The results on attacking more surrogates by RAD are reported in Table 8.

Table 7: Detection mAP in different hyper-parameters in RAD

| Strategy | Parameter | M1 | M2 | M3 | M4 | M5 | M6 | M7 | M8 | M9 |
|---|---|---|---|---|---|---|---|---|---|---|
| Dynamic | -1 | 18.3 | 1.2 | 20.0 | 20.6 | 24.3 | 22.8 | 26.7 | 28.3 | 40.0 |
| | -2 | 18.2 | 1.2 | 20.1 | 20.7 | 24.2 | 22.8 | 26.6 | 28.0 | 39.8 |
| | -3 | 18.4 | 1.3 | 20.3 | 20.8 | 24.3 | 22.9 | 26.7 | 28.5 | 40.2 |
| Static | 10 | 18.2 | 1.1 | 19.9 | 20.5 | 24.2 | 22.8 | 26.2 | 28.0 | 39.8 |
| | 20 | 18.1 | 1.2 | 19.9 | 20.5 | 24.3 | 22.6 | 26.4 | 28.2 | 39.9 |
| | 30 | 18.2 | 1.3 | 20.1 | 20.7 | 24.3 | 22.8 | 26.0 | 27.9 | 40.1 |

Table 8: Detection mAP of RAD on different surrogates

| Surrogate | M1 | M2 | M3 | M4 | M5 | M6 | M7 | M8 | M9 |
|---|---|---|---|---|---|---|---|---|---|
| YOLOv3 | 14.6 | 0.7 | 16.3 | 17.0 | 20.4 | 19.1 | 22.3 | 23.8 | 35.0 |
| Mask R-CNN | 20.4 | 24.2 | 25.7 | 26.5 | 1.1 | 28.9 | 33.1 | 34.9 | 45.4 |
| RetinaNet | 20.7 | 6.1 | 2.3 | 25.7 | 29.3 | 28.2 | 31.7 | 33.8 | 44.1 |

# E    IMPLEMENTATION DETAILS

## E.1    PRE-PROCESSING

To pre-process, we resize the image with its long side as 416 for YOLOv3 or RetinaNet and 448 for Mask R-CNN, and then zero-pad it to a square. The resolution is kept relatively the same for a fair evaluation. Images are normalized to [0,1] in YOLOv3 or subtracted by the mean of COCO training set in RetinaNet and Mask R-CNN. Accordingly, samples in AOCO detection have the long side 416 and that for AOCO segmentation is 448.

## E.2    TRANSFER-ENHANCING UPDATE TECHNIQUES

DI (Xie et al. (2019)) transforms the image for 4 times with probability $p$ ($p = 1$ for better transferability as suggested) and averaging the gradients. The transformation is to resize the image to $0.9\times$ its size and randomly padding the outer areas with white pixels. SI (Lin et al. (2020)) divides the sample numerically by the power 2 for 4 times and averages the 4 obtained gradients. TI (Dong et al. (2019)) translates the image to calculate the augmented gradients. To implement it efficiently, it adopts a kernel to simulate the averaging of gradients. We choose the kernel size 15 as suggested. MI (Dong et al. (2018)) uses momentum optimization (parameter $\mu = 1$ as suggested) for a better transferability and a faster attack. Cross-domain attack (Naseer et al. (2019)) uses extra datasets (paintings, denoted as CD-paintings, and comics, denoted as CD-comics) to train a perturbation generator with the relative loss. The adopted surrogate model is also InceptionV3 for consistency. All perturbations are resized to fit the sample size.

## E.3    DETECTION ATTACKS

For DAG (Xie et al. (2017)), we follow the setting of generating dense proposals. The classification probabilities of 3000 bounding boxes with highest confidence are attacked. But we alter its optimization to (5) because its original update produces quite small perturbation, leading to a poor transferability, which is unfair for comparison. Dfool (Lu et al. (2017)) suppresses the classification confidence for the original bounding boxes, which is the same in our experiment. Localization loss is shown to be useful in (Zhang & Wang (2019)), and here we suppress the width and height of the original bounding boxes.

# F    VISUAL RESULTS OF RAD PROCESS

By RAD, the relevance map is attacked to be meaningless and loss its focus. In Fig. 8, the initial prediction is correct and the relevance map is clear. RAD constantly misleads the relevance map to be unstructured without outline of objects. Finally, all bounding boxes vanish.

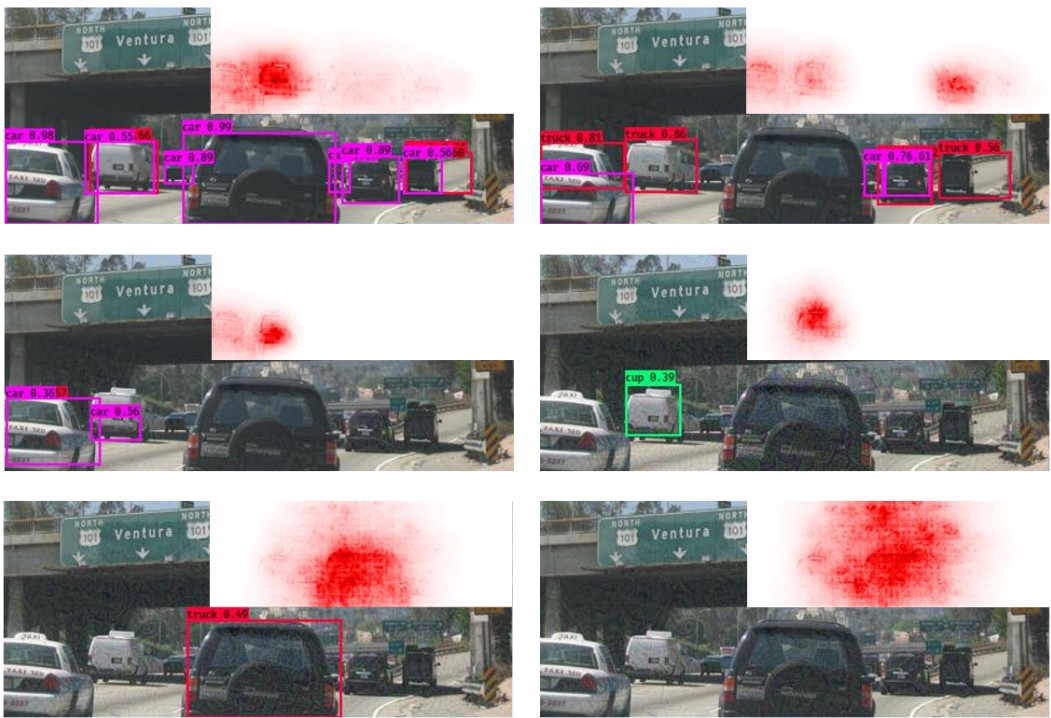

Figure 8: Transition of prediction and relevance map in RAD (from top to bottom and left to right).

## G  MORE ABOUT AOCO

We report the mAP50 and mAP75 of AOCO in Table 9 and Table 10. We show in Fig. 9 the visual comparative results.

Table 9: Detection mAP50 and segmentation mAP50 on COCO and AOCO

|                   | M1   | M2   | M3   | M4   | M5   | M6   | M7   | M8   | M9   |
|-------------------|------|------|------|------|------|------|------|------|------|
| COCO detection    | 49.2 | 56.4 | 58.1 | 62.0 | 63.8 | 60.7 | 64.1 | 66.0 | 74.3 |
| AOCO detection    | 26.7 | 1.6  | 27.6 | 29.1 | 34.5 | 29.9 | 34.3 | 37.4 | 51.7 |
| COCO segmentation | \    | \    | \    | \    | 60.6 | \    | 61.3 | 63.3 | \    |
| AOCO segmentation | \    | \    | \    | \    | 2.4  | \    | 17.9 | 18.9 | \    |

Table 10: Detection mAP75 and segmentation mAP75 on COCO and AOCO

|                   | M1   | M2   | M3   | M4   | M5   | M6   | M7   | M8   | M9   |
|-------------------|------|------|------|------|------|------|------|------|------|
| COCO detection    | 30.8 | 35.8 | 40.6 | 44.6 | 46.3 | 46.3 | 50.0 | 51.2 | 59.9 |
| AOCO detection    | 14.2 | 0.6  | 16.5 | 17.1 | 20.8 | 19.9 | 23.3 | 24.6 | 37.4 |
| COCO segmentation | \    | \    | \    | \    | 40.9 | \    | 42.9 | 44.1 | \    |
| AOCO segmentation | \    | \    | \    | \    | 1.0  | \    | 11.9 | 12.6 | \    |

COCO detection      AOCO detection      COCO segmentation      AOCO segmentation

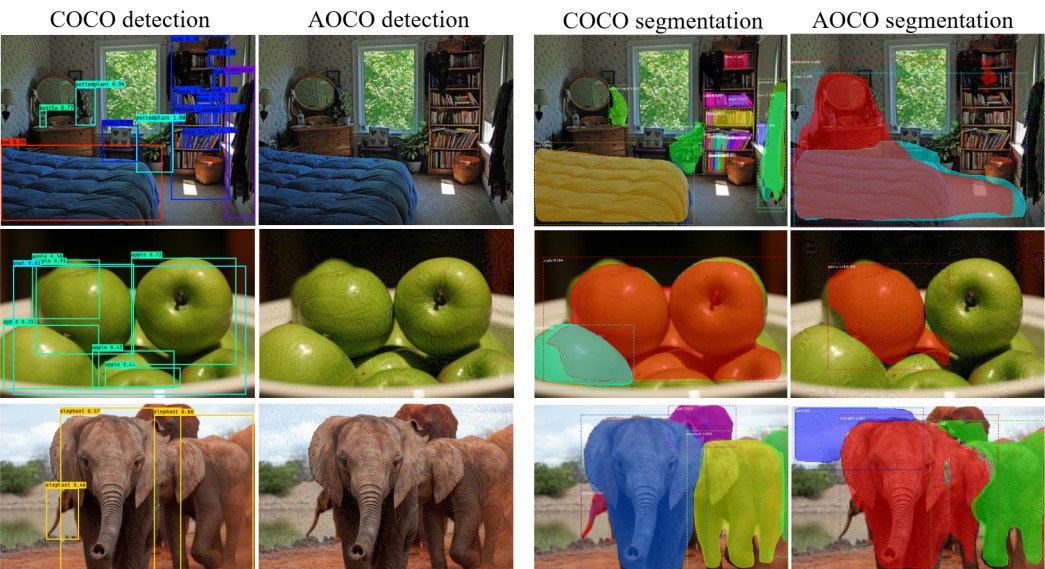

Figure 9: Detection and segmentation results in COCO and AOCO by YOLOv3 and Mask R-CNN. For COCO, both networks predict correctly. For AOCO segmentation results, the top image contains two big masks for "chair" and "potted plan"; the second image contains one false mask for "sports ball"; the bottom image contains "dog" in green, "car" in purple and "elephant" in red.

