# OpenReview forum: "Relevance Attack on Detectors"
_ICLR.cc/2021/Conference — Reject_

### Official Review · AnonReviewer2 · 2020-10-25
**A relevance map proposed for adversarial attack on object detection. The overall contribution seems limited.**

**Rating:** 4
**Confidence:** 4

**Review:**

A transferable adversarial attack method is proposed for object detection. A relevance map is used to discover the common weakness of existing detectors. An adversarial dataset for object detection is created for experimental validation.

The relevance map proposed indeed derives from SGLRP as mentioned in Sec. 3.3. The only modification is to change a single node target t to the average of a set of nodes.  As the major contribution claimed is on the relevance map, the minor modification makes the contribution limited.

The relevance map is computed by back-propagating the relevance R from the final layer to the input following rules as illustrated in Sec. 3.3. However, the proposed method is claimed to perform a black-box transferable attack. Also, the gradients are utilized to update the relevance map. It is thus not clear how this relevance map correlates to the black-box transferable attack.

In the experiments, the proposed attack method shall be compared to sota detection or segmentation-based methods (e.g., EfficientDet, Cascade-RCNN, Libra-RCNN).

---

> ### Author Response · Authors · 2020-11-19
> **Response to Reviewer 2**
>
> We would like to thank reviewer 2 for the thoughtful comments and efforts towards improving our manuscript.
>
>
> Question 1. The major contribution is modifying SGLRP, which is limited.
>
> Answer 1. We sincerely appreciate this constructive review. RAD is a complete attack framework, including the design of Multi-Node SGLRP, a proper choice of target nodes, and a suitable update approach. Simply extending SGLRP to Multi-Node SGLRP does not work as empirically validated in Table 1. In-depth analysis and trial also contribute to RAD’s great performance.
>
>
> Question 2. It is unclear why gradients from the relevance maps are required in black-box attack.
>
> Answer 2. We are grateful to the reviewer for pointing out the vagueness. In transfer-based black-box attack, one attacks a surrogate model in a white-box manner and sends the adversarial samples to attack black boxes. Thus, the gradients of the white-box model (and its relevance maps in RAD) are required.
>
>
> Question 3. Authors are expected to attack more SOTA black-box models.
>
> Answer 3. Thanks for this thoughtful review. We have attacked Cascade R-CNN (M6) in the previous submission, and the performance of EfficientDet (M9) is reported in all tables in the updated submission.

---

### Official Review · AnonReviewer4 · 2020-10-28
**The Topic is not very valuable**

**Rating:** 5
**Confidence:** 3

**Review:**

Summary:
The paper found a new way, called Relevance Attack on Detectors (RAD),  to generate high transferability adversarial examples against detectors by suppressing the multi-node relevance.
Moreover, a dataset generated by this attacking method is introduced.

Pros:
1. The designed RAD is new and technically sound.
2. The experimental results on multiple detectors, like YOLOv3,  RetinaNet, Mask R-CNN, etc. are promising.

Cons:
1. It's not surprising that the object detectors can be attacked together.  One naive way is that using the ensemble attack by average the negative CE loss like the paper "Making an Invisibility Cloak: Real World Adversarial Attacks on Object Detectors" proposed. So I think some baselines need to be added in experiments.
2.  By imposing diversity transformation on input image can also improve transferability ("Boosting Adversarial Attacks with Momentum"). Similar rules also can be applied in detectors.
3. I cannot see a very significant value of the generated dataset. Like in the image classification dataset, one can generate adversarial examples easily, but there is not an individual dataset that only contains generated adversarial examples generated by one specific method. Some famous dataset like imagenet-C ("Benchmarking Neural Network Robustness to Common Corruptions and Perturbations") is designed by natural perturbations.

---

> ### Author Response · Authors · 2020-11-19
> **Response to Reviewer 4**
>
> We would like to thank reviewer 4 for the thoughtful comments and efforts towards improving our manuscript.
>
>
> Question 1. More baselines are necessary.
>
> Answer 1. We sincerely appreciate this constructive review. Since we received the review, we have been trying to conduct more comparative experiments. However, due to the time limitation, we cannot report the results at this moment. We thank the reviewer for pointing this related work, which has been discussed in the updated submission in the second paragraph of Section 2.
>
>
> Question 2. Update approaches for transferability in attacking classifiers could be adopted.
>
> Answer 2. Thanks for pointing out that. Actually, related discussions on update approaches are in Section 3.5, e.g., MI stands for momentum update and DI is for input diversity. Empirically, SI is most suitable for RAD and therefore chosen.
>
>
> Question 3. The value of dataset is unclear.
>
> Answer 3. We would like to thank reviewer 4 for the thoughtful comments. There is no dataset containing adversarial samples for detectors only, not because the dataset is useless, but because existing methods are not capable of conducting high-transferable attack and the resulting samples are detector-dependent. Given that RAD achieves high transferability, we could provide such an adversarial dataset. With the dataset, RAD benefits the community very directly, because designers could easily use AOCO to evaluate and improve the robustness of detectors.

---

### Official Review · AnonReviewer1 · 2020-10-29
**large room for improvement**

**Rating:** 4
**Confidence:** 3

**Review:**

**Summary**:
This work proposes to attack object detectors by targeting their relevance maps of the different detected objects. The proposed RAD attack shows better black box transferability across different detectors on MSCOCO dataset. The relevance maps are calculated based on SGLRP act as an attention mechanism to the attack to focus on relevant regions in the more meaningful image and hence produce more transferable attacks.

**Strengths** :
- Good attack performance and transferability between detectors, which poses a security threat for SDV applications that use object detectors
- Eight different detectors and three segmentation models are used in the RAD attack, which shows good generalization.

**Weaknesses**:
- Missing important references [a,b,c]. All of these works attack object detectors and target transferability.
- The paper is poorly written and ambiguous. Variables are introduced without proper definitions. It is not clear how to obtain the gradients in eq(3) with respect to the relevance maps.
- No use of the proposed dataset. The authors propose a new dataset of adversarial objects but never mention or showcase the dataset's usefulness. A straightforward way to show the dataset's usefulness is by performing adversarial training and making robust detectors against the proposed attacks.
- No enough ablation is performed. The only ablation to the proposed method is in table 7 regarding the way to pick the detection target. The relevance maps based on LRP are expensive and worse than recent saliency maps like CAM and grad-CAM. The attack budget $\epsilon =16$ picked in the experiments is not justified ( it might be big or small for attack success ), and a plot of mAP vs. $\epsilon$ for different detectors would give more information about the effect of the attack.
- All the attacks in the paper are performed on YOLOv3 and transferred to other models. It would be more informative to show transferability matrices of attacks performed on all models and transferred to all others.
- The novelty of the proposed methodology is limited. While the use of relevance maps to improve the transferability of attacks on object detectors is novel, no proper explanation is provided. The attacks are based on PGD, and the relevance map is adapted from SGLRP. The paper offers no theoretical results or exciting insights.



[a] Huang et al. "Universal Physical Camouflage Attacks on Object Detectors", ( CVPR 2020)
[b] Wu et al. "Making an Invisibility Cloak: Real World Adversarial Attacks on Object Detectors", (ECCV 2020 )
[c] Xu et al. "Adversarial T-shirt! Evading Person Detectors in A Physical World" (ECCV 2020).


Minor issues :
- Many grammar mistakes:" because they possess multiple-output.", "Among the classification attacks and detection ones, cross-domain attack (Naseer et al. (2019)) is the most effective, but RAD is more aggressive" ..etc.
- No question marks in titles 3.1-3.5.
- Table 2-5 could have been visualized better by using a bar chart, for example, to observe the relative performance of attacks and defenses.

---

> ### Author Response · Authors · 2020-11-19
> **Response to Reviewer1**
>
> We would like to thank reviewer 1 for the thoughtful comments and efforts towards improving our manuscript.
>
>
> Question 1. Authors should refer to [a, b, c].
>
> Answer 1. We thank the reviewer for pointing these related works. These papers are referred in the updated submission in the second paragraph of Section 2. They conduct white-box attack in physical world. In contrast, we conduct black-box attack on digital images. Accordingly, they target at transferability across scenes, and we aim at transferability across models. Our goals differ in essence.
>
>
> Question 2. It is not clear how to obtain the attack gradients.
>
> Answer 2. Thanks for pointing out the vagueness. The calculation of the attack gradients has been described in the updated submission in Section 3.3.
>
>
> Question 3. The dataset is not useful.
>
> Answer 3. AOCO dataset serves as a potential benchmark to evaluate the robustness of detectors, which will be beneficial to network designers. It will also be useful for adversarial training, as the most effective practice to clearly improve the robustness of DNNs [d, e, f]. Notice that there is no other adversarial dataset for detection at all. This is not because the dataset is useless, but due to the low transferability of attack methods such that the examples are detector-dependent. Now we have achieved high transferability and can then make such an adversarial dataset publicly available.
>
>
> Question 4. The ablation study is not sufficient.
>
> Answer 4. We sincerely appreciate this constructive review. Below we explain why we choose SGLRP and the attack budget. SGLRP [g] is of the latest visualization methods compared to Grad-CAM [h], which, if adopted in RAD, requires the time-consuming second-order gradients and limits the usage. Besides, SGLRP excels in discriminating ability against irrelevant regions of a certain target node. Accordingly, SGLRP is adopted in our RAD. The attack budget follows the setting in [i, j, k], where 16 is validated to be an ideal threshold for the trade-off between effectiveness and imperceptibility in attack. Besides hyperparameters, we also conduct ablation study to add large random noise. Results in Table 3 show that the transferability comes from attack rather than large perturbations.
>
>
> Question 5. It would be more informative to attack other surrogate models.
>
> Answer 5. Thanks to your constructive suggestions, we have added the results of attacking Mask-RCNN and RetinaNet in the updated submission in Appendix D.
>
>
> Question 6. Authors provide no proper explanation, limiting the novelty of RAD.
>
> Answer 6. We are grateful to reviewer 1 for the thoughtful comments. To the best of our knowledge, we are the first to propose that one plausible way to improve the transferability of detector attacks is to find common properties across them, which facilitates the discovery of common weaknesses. Based on them, the designed attack can threaten variable victims. RAD focuses on relevance maps, and attacking other common properties might also be possible, which is an implicit contribution of RAD. Indeed, the above is only some heuristic guess. Providing convincing theoretical explanations is hard but will certainly be our next focus.
>
> Question 7. The presentation could be improved.
>
> Answer 7. Thanks for your careful reading. The presentation of paper has been improved in the updated submission.
>
>
> [d] Mitigating adversarial effects through randomization, ICLR 2018.
>
> [e] Ensemble adversarial training: Attacks and defenses, ICLR 2018.
>
> [f] Adversarial training for free!, NeurIPS 2019.
>
> [g] Explaining convolutional neural networks using softmax gradient layer-wise relevance propagation, ICCV Workshop, 2019.
>
> [h] Grad-CAM: Visual Explanations from Deep Networks via Gradient-based Localization, CVPR 2017.
>
> [i] Evading defenses to transferable adversarial examples by translation-invariant attacks, CVPR 2019.
>
> [j] Improving transferability of adversarial examples with input diversity, CVPR 2019.
>
> [k] Nesterov accelerated gradient and scale invariance for adversarial attacks, ICLR 2020.

---

> > ### Comment · AnonReviewer1 · 2020-11-20
> > **more issues to be resolved**
> >
> > I would like to thank the authors for providing a good rebuttal for my concerns. Here are my comments.
> >
> > *Answer 2. Thanks for pointing out the vagueness. The calculation of the attack gradients has been described in the updated submission in Section 3.3.*
> >
> > Comment 2. I still do not understand how Eq. 3 is used in calculating the attack gradients in Eq 4. It is mentioned that iNNvestigate library is used, but that is not an explanation of **how** the gradients are computed with respect to SGLRP relevant maps ( which itself is an iterative process). Are second-order gradients used (gradients of gradients )?
> >
> > *Answer 3. AOCO dataset serves as a potential benchmark to evaluate the robustness of detectors, which will be beneficial to network designers. It will also be useful for adversarial training, as the most effective practice to clearly improve the robustness of DNNs [d, e, f]. Notice that there is no other adversarial dataset for detection at all. This is not because the dataset is useless, but due to the low transferability of attack methods such that the examples are detector-dependent. Now we have achieved high transferability and can then make such an adversarial dataset publicly available.*
> >
> > Comment 3. The dataset is useful in essence, but the authors did not showcase its usefulness. Will training MaskRCNN on this proposed dataset improve its robustness to the proposed RAD attacks and all other attacks?
> >
> > *Answer 4. We sincerely appreciate this constructive review. Below we explain why we choose SGLRP and the attack budget. SGLRP [g] is of the latest visualization methods compared to Grad-CAM [h], which requires the DNN to have a specific structure and limits the usage. Besides, SGLRP excels in discriminating ability against irrelevant regions of a certain target node. Accordingly, SGLRP is adopted in our RAD. The attack budget follows the setting in [i, j, k], where 16 is validated to be an ideal threshold for the trade-off between effectiveness and imperceptibility in the attack. Besides hyperparameters, we also conduct ablation study to add large random noise. Results in Table 3 show that the transferability comes from attack rather than large perturbations.*
> >
> > comment 4. I respectfully disagree with the authors that $\epsilon = 16$ is an *ideal* threshold. Just because some papers conveniently used it on ImageNet classifiers doesn't imply it is best to use it on MSCOCO detectors. In fact, the formal and proper way to present attack results is to show the entire plot of performance (mAP in this case) vs the attack budget ( $\epsilon$) as pointed out by the pioneers of adversarial attacks ( Carlini,  Madry, Goodfellow, and others ) in their seminal work [a] .
> >
> > [a]  "On Evaluating Adversarial Robustness". Nicholas Carlini, Anish Athalye, Nicolas Papernot, Wieland Brendel, Jonas Rauber, Dimitris Tsipras, Ian Goodfellow, Aleksander Madry, Alexey Kurakin;  Arxive 2019

---

> > > ### Author Response · Authors · 2020-11-23
> > > **Further Response to Reviewer1**
> > >
> > > We sincerely thank reviewer 1 for the additional comments and efforts towards improving our work. According to your insightful comments, we have revised our submission again. Below is our further response.
> > >
> > >
> > > Comment 2. I still do not understand how Eq. 3 is used in calculating the attack gradients in Eq 4. It is mentioned that iNNvestigate library is used, but that is not an explanation of how the gradients are computed with respect to SGLRP relevant maps (which itself is an iterative process). Are second-order gradients used (gradients of gradients)?
> > >
> > > Answer 2. We are grateful to this constructive comment. The calculation of the attack gradients has been described in the updated submission in Appendix A, where we specify the rules of calculating relevance maps, which naturally result in the gradients. Actually, SGLRP back-propagates the relevance only ONCE with the certain specified rules, so it is not a computation-consuming iterative process. Also, according to the rules, RAD requires no costly calculation of second-order gradients, which is needed if CAM is adopted.
> > >
> > >
> > > Comment 3. The dataset is useful in essence, but the authors did not showcase its usefulness. Will training Mask RCNN on this proposed dataset improve its robustness to the proposed RAD attacks and all other attacks?
> > >
> > > Answer 3. We very much appreciate reviewer 1 for the thoughtful comments towards experimental design. Since we received the review, we have been trying to conduct adversarial training. However, due to the time limitation, we cannot report the results at this moment. Actually, good adversarial training requires specific researches, for which the dataset will be always a basis. As an analogy, there are many advanced methods for adversarial training for classification tasks [e, f].
> > >
> > >
> > > Comment 4. I respectfully disagree with the authors that is an ideal threshold. Just because some papers conveniently used it on ImageNet classifiers doesn't imply it is best to use it on MSCOCO detectors. In fact, the formal and proper way to present attack results is to show the entire plot of performance (mAP in this case) vs the attack budget as pointed out by the pioneers of adversarial attacks (Carlini, Madry, Goodfellow, and others) in their seminal work (“On Evaluating Adversarial Robustness”).
> > >
> > > Answer 4. Thanks to your detailed comments. The epsilon setting was following published literature [i, j, k] but we also respect the reviewer’s opinion that discussing the influences of the attack budget is important. After receiving this comment, we immediately conduct ablation study on the transferability when RAD adopts an attack budget of 4, 8, 12, 16, 20. The corresponding results have been reported in Figure 7 in the updated submission.

---

### Official Review · AnonReviewer3 · 2020-11-01
**interesting idea with good transferrability**

**Rating:** 6
**Confidence:** 4

**Review:**

This paper presents a method for adversarial attacks on object detectors by exploiting relevance maps that are originally intended for model interpretation. Unlike most of the existing methods that attack detection scores directly, the proposed approach focuses on suppressing the relevance map associated with target objects by image perturbation. The idea is interesting and demonstrates good transferability on the tasks of object detection and segmentation.

The paper is mostly well written and easy to follow. The adversarial object dataset can also be helpful to the research community.

The main downside of the paper is that some of the comparisons are not apple-to-apple in the experiments. For example, the proposed approach applies update techniques (i.e. Translation-Invariant) to improve transferability. However, it is not my impression that this was done on the baseline methods, which leads to unfair comparison.

Some technical details need to be better articulated in the paper. For example, there is no mentioning of the adversarial loss function and how it is optimized. As pointed out in [2], attacking CNN interpretations is not trivial. Without the details of how to update the gradients of the relevance map, It would make reproducibility difficult.

While focusing on a different problem,  the proposed approach shares some similarities with methods designed to attack model interpretations such as [1] and [2], which should be discussed as related work.

[1] Amirata Ghorbani, Abubakar Abid, and James Zou. Interpretation of neural networks is fragile. In Proceedings of the AAAI Conference on Artificial Intelligence, volume 33, pp. 3681–3688, 2019
[2] Xinyang Zhang, Ningfei Wang  Hua Shen, Shouling Ji, Xiapu Luo,Ting Wang, Interpretable Deep Learning under Fire, USENIX Security '20

---

> ### Author Response · Authors · 2020-11-19
> **Response to Reviewer3**
>
> We would like to thank reviewer 3 for the thoughtful comments and efforts towards improving our manuscript.
>
>
> Question 1. Unfair comparison may occur when update techniques are not applied to baselines.
>
> Answer 1. We very much appreciate reviewer 3 for the thoughtful comments towards experimental design. It is notable that without update techniques, RAD also outperforms baselines as in Tables 2 and 3. Accordingly, we believe that advantages of RAD are validated fairly. We do not combine these two tables because the choice of update technique is a part of the RAD design presented in method part. Transferable update techniques are adopted in PGD baselines in Table 3.
>
>
> Question 2. The adversarial loss and corresponding gradients for optimization is not clear.
>
> Answer 2. We are grateful to this constructive review. The adversarial loss is the relevance map h(x, T) as in (3). The calculation of the attack gradients has been described in the updated submission in Appendix A for reproductivity. Also, we attach the source code as supplementary materials.
>
>
> Question 3. The difference between RAD and [1, 2] should be discussed.
>
> Answer 3. We thank the reviewer for pointing out these related works. The difference between RAD and these works are listed below and added to the updated submission in the last paragraph of Section 2. RAD misleads detectors by suppressing relevance maps. In contrast, [1] misleads the relevance maps while keeping the prediction unchanged. [2] also misleads DNNs, but it keeps the relevance maps unchanged.

---

### Author Response · Authors · 2020-11-23
**Summary of changes in the updated submission**

Dear Program Chairs, Area Chairs, and Reviewers,

First of all, we would like to thank you for your time, constructive critiques, and valuable suggestions. Your input contributed to a significant improvement of the paper and to our proposed method as well. We did our best to address your comments on our revised manuscript. For each reviewer, we give a detailed response to each of your comments. Major changes made to the previously submitted document are summarized below for you.
* The calculation of attack gradients from the relevance maps is described in detail in Appendix A.
* More black-boxes (EfficientDet, M9, see Tables) and white-boxes (Mask R-CNN, RetinaNet, see Appendix D) are attacked.
* The value of AOCO dataset is clarified in Section 5.
* Important related works are discussed in Section 2.

Sincerely yours,

ICLR 2021 Paper248 Authors

---

### Decision · Program_Chairs · 2021-01-07
**Final Decision**

**Decision:**

Reject

**Comment:**

This paper proposes a transferable adversarial attack method for object detection by using the relevance map. Four reviewers provided detailed reviews: 2 of them rated “Ok but not good enough - rejection”, 1 rated “Marginally below” and 1 rated “Marginally above”. While reviewers consider the paper well written and using relevance map novel, a number of concerns are raised, including limited novelty, the lack of theoretical results, no use of the proposed dataset, insufficient ablation, etc. During the rebuttal, the authors made efforts to response to all reviewers’ comments. However, the major concerns remain, and the rating were not changed. The ACs concur these major concerns and agree that the paper can not be accepted at its current state.